# Lipidomic Profiling and Storage-Induced Changes in Cassava Flour Using LC-MS/MS

**DOI:** 10.3390/foods13193039

**Published:** 2024-09-25

**Authors:** Peixu Du, Qinfei Wang, Yi He, Houmei Yu, Liming Lin, Zhenwen Zhang

**Affiliations:** 1National R&D Centre for Potato Processing/Tropical Crops Genetic Resources Institute, China Academy of Tropical Agriculture Science, Haikou 571101, China; 18768667346@163.com (P.D.); wangqf508@163.com (Q.W.); yuhoumei@sina.com (H.Y.); liminglin2010@126.com (L.L.); 2Key Laboratory of Ministry of Agriculture for Germplasm Resources Conservation and Utilization of Cassava, Haikou 571101, China; 3Key Laboratory for Deep Processing of Major Grain and Oil, Ministry of Education, Hubei Key Laboratory for Processing and Transformation of Agricultural Products, School of Food Science and Engineering, Wuhan Polytechnic University, Wuhan 430023, China; 4National R&D Center for Se-rich Agricultural Products Processing, Hubei Engineering Research Center for Deep Processing of Green Se-rich Agricultural Products, School of Modern Industry for Selenium Science and Engineering, Wuhan Polytechnic University, Wuhan 430023, China; yi.he@whpu.edu.cn

**Keywords:** cassava flour, lipidomics profile, storage, LC-MS/MS

## Abstract

Cassava serves as a primary staple food for over one billion people worldwide. The quality of cassava flour is markedly affected by the oxidation and deterioration of lipids during storage. Despite its significance, the lipid composition of cassava flour and its alterations throughout storage periods have not been extensively studied. This study offers a comprehensive lipidomic analysis of cassava flour over storage periods using liquid chromatography–mass spectrometry/mass spectrometry (LC-MS/MS). The results showed that 545 lipids from five classes and 27 subclasses were identified in cassava flour, including key substances such as free fatty acids (36 species), diglycerides (DGs) (31 species), and triglycerides (TGs) (259 species). Using Metware Cloud for statistical analysis, significant variations were observed in 50 lipid species over long-term storage, reflecting changes in lipid profiles due to storage. These lipids correlate with seven metabolic pathways, among which glycerolipid metabolism is the most affected. The metabolites associated with these pathways can differentiate cassava flour based on the length of storage. This study provides a theoretical basis and storage technology parameters for lipid changes during cassava flour storage.

## 1. Introduction

Cassava (*Manihot esculenta* Crantz) is considered one of the most crucial tuber crops in the tropical and subtropical regions, providing a primary food source for over one billion people globally [1]. However, its utilization is hampered by its short shelf-life due to post-harvest physiological deterioration (PPD) [2]. Consequently, fresh cassava roots must be quickly processed into a shelf-stable dry product to mitigate the effects of PPD, while processing cassava roots into flour also improves shelf-life stability [3]. In addition, cassava flour itself does not contain gluten, making it an excellent choice for baking and cooking [4,5].

Previous studies have shown that the storage quality of edible cassava flour is closely related to a number of factors, among which fatty acids are the factors that have a greater impact on the storage characteristics, similar to pasta products [6]. Generally speaking, the safe shelf-life of cassava flour is one year, but after long-term storage for 180 days, the flavor will change and unsaturated fatty acids will decrease significantly [7]. Although the lipid content in cassava flour is relatively low, about 2.5%, fatty acid degradation also occurs during storage, but other lipid changes related to rancidity are not clear [8].

Lipids play a crucial role in cassava, forming the basis of sensory, biochemical, and physiological events [9]. Lipids are a diverse group of compounds, characterized by their insolubility in water and solubility in nonpolar organic solvents, encompassing fatty acids, sterols, and phospholipids. They are commonly classified into eight groups: fatty acids, glycerolipids, glycerophospholipids, sphingolipids, sterol lipids, prenol lipids, saccharolipids, and polyketides [10,11]. According to the Lipid Maps database, over 43,300 unique lipid structures are cataloged, including regional isomers, oxidized lipids, and other modified forms [12]. Lipids also affect the quality of food during storage. Studies have shown that rice bran and brown rice are prone to rancidity due to lipid hydrolysis and oxidation during storage, and phospholipid and glycerolipid metabolism are the key metabolic pathways involved, among which the degradation of fatty acids plays a decisive role [13,14]. Understanding lipid composition and rancidity can provide subsequent technical support for food storage. For example, a microcalorimetry study can control the oxidation of linoleic acid and rancidity for food storage [15]. Cassava also causes lipid oxidation product changes due to PPD [16]. However, studies and the mechanisms of rancidity due to lipid changes in cassava flour during storage have rarely been reported.

Lipidomics, a subfield of metabolomics proposed in 2003, focuses on the comprehensive study of lipids within biological systems [17]. Lipidomics, which is largely based on mass spectrometry, involves systematic and large-scale studies of the structure, composition, and quantity of lipids in biological systems such as organs, cells, and body fluids [18]. Chromatography coupled with mass spectrometry, especially liquid chromatography–tandem mass spectrometry (LC-MS/MS), enables the accurate characterization and quantification of lipids [19]. Lipidomics methods are now widely used in various research fields [20,21]; using this method to analyze and understand the lipid composition and storage changes of cassava flour is crucial for predicting the storage quality of cassava flour and providing guidance on storage methods.

‘South China No.9’ (SC9) and ‘GuiRe No.10’ (GR10) are two representative edible cassava varieties in China, widely grown in Guangxi and Hainan. This study aims to analyze the lipidomic characteristics of cassava flour from these two varieties and their changes during storage using LC-ESI-MS/MS mass spectrometry. First, we used QTRAP^®^ 6500+ UPLC-MS/MS to perform lipidomic analysis on fresh (day 0) and stored cassava flour (day 180 and day 540) [22], and then, we used the Metware Database (MWDB) based on retention time (RT) and daughter ion pair information to identify compound lipids. LDGTS, TG, MG, Cert, FFA, CoQ, PI, and PS in cassava flour are quantified by the multiple reaction monitoring (MRM) mode of triple quadrupole mass spectrometry. Finally, the lipidomic characteristics of these two cassava varieties and their changes during storage were systematically compared and analyzed. This was to explore the intrinsic causes of lipid rancidity in different varieties of edible cassava flour during storage.

## 2. Materials and Methods

### 2.1. Reagents

The roots of cassava varieties ‘South China No.9’ (SC9) and ‘GuiRe No.10’ (GR10) were provided by the National Cassava Germplasm Repository, which were harvested in the field (19°30′33.13″ N, 109°30′19.34″ E) of Danzhou City, Hainan Province, China, in April 2022. HPLC-grade acetonitrile (ACN), methanol (MeOH), isopropanol (IPA), dichloromethane (CH_2_Cl_2_), and methyl tert-butyl ether (MTBE) were purchased from Merck (Darmstadt, Germany). HPLC-grade formic acid (FA) and ammonium formate (AmFA) were obtained from Sigma-Aldrich (St. Louis, MO, USA). Ultrapure water was obtained using a Milli-Q system (Millipore, Billerica, MA, USA). Lipid standards were purchased from Sigma-Aldrich or Avanti Polar Lipids (Alabaster, AL, USA).

### 2.2. Sample Preparation and Extraction

Cassava roots were peeled, cleaned, dried at 50 °C for 24 h in an oven from Sigma-Aldrich (St. Louis, MO, USA), and then ground into flour by a high-speed multi-function pulveriser (Bao’o Electric Appliance Co., Hangzhou, China) after using an 80-mesh sieve (Bioroyee Biotechnology Co., Beijing, China). The flour was sealed in Ziploc bags (food grade) and stored in a sample library at room temperature (25–30 °C) with humidity controlled at 60% ± 5% to simulate natural conditions. The lipid extraction procedure was based on the method by [23] with minor modification. Specifically, 20 mg of dry sample was placed in a 2 mL centrifuge tube containing one steel bead (internal diameter approximately 4 mm). A solvent mixture of MTBE and MeOH (3:1) was added, followed by vortexing for 30 min. Then, 300 μL of ultrapure water was added, and the sample was vortexed for an additional minute before being left at 4 °C for 10 min. The sample was centrifuged at 12,000 r/min for 3 min at 4 °C, and 400 μL of the supernatant was transferred to a 1.5 mL centrifuge tube and concentrated until completely dry at 20 °C. Finally, 200 μL of a lipid complex solution (ACN: IPA = 1:1) was added to redissolve the sample, followed by vortexing for 3 min and centrifuging for 3 min at 12,000 r/min at 4 °C. A total of 120 μL of the reconstituted solution was collected for LC-MS/MS analysis.

### 2.3. HPLC Conditions

The sample extracts were analyzed using an LC-ESI-MS/MS system (UPLC, ExionLC AD, https://sciex.com.cn/; MS, QTRAP^®^ 6500+ System, https://sciex.com) accessed on 1 November 2023. The lipid extracts of SC9 and GR10 cassava flour were separated by positive and negative ionization using a Thermo Accucore™ C30 column (2.6 μm, 2.1 mm × 100 mm i.d.). A binary solvent system consisted of mobile phase A (acetonitrile/water, 60:40, *v*/*v*) containing 0.1% formic acid and 10 mmol/L ammonium formate, and mobile phase B (acetonitrile/isopropanol, 90:10, *v*/*v*) containing 0.1% formic acid and 10 mmol/L ammonium formate. Gradient elution was performed at a flow rate of 0.35 mL/min at 45 °C with an injection volume of 2 μL. The gradient program was as follows: A/B (80:20, *v*/*v*) at 0 min, 70:30 *v*/*v* at 2.0 min, 40:60 *v*/*v* at 4 min, 15:85 *v*/*v* at 9 min, 10:90 *v*/*v* at 14 min, 5:95 *v*/*v* at 15.5 min, 5:95 *v*/*v* at 17.3 min, 80:20 *v*/*v* at 17.3 min, and 80:20 *v*/*v* at 20 min. The effluent was alternately connected to the ESI-Triple Quadrupole Linear Ion Trap (QTRAP)-MS. The analyzed conditions refer to the relevant literatures [24,25].

### 2.4. ESI-MS/MS Conditions

LIT and triple quadrupole (QQQ) scans were acquired on a triple quadrupole-linear ion trap mass spectrometer (QTRAP) QTRAP^®^ 6500+ LC-MS/MS System, equipped with an ESI Turbo Ion-Spray interface, operating in positive and negative ion modes, and controlled by Analyst 1.6.3 software (Sciex). The ESI source operation parameters were as follows: ion source, turbo spray; source temperature, 500 °C; ion spray voltage (IS): 5500 V (Positive), −4500 V (Negative); ion source gas 1 (GS1), gas 2 (GS2), and curtain gas (CUR) were set at 45, 55, and 35 psi, respectively. Instrument tuning and mass calibration were performed with 10 and 100 μmol/L polypropylene glycol solutions in QQQ and LIT modes, respectively. QQQ scans were acquired as MRM experiments with collision gas (nitrogen) set to 5 psi. DP and CE for individual MRM transitions were optimized further. A specific set of MRM transitions was monitored for each period according to the metabolites eluted within this period [26,27].

### 2.5. Lipid Qualitative and Quantitative Principle

Qualitative analysis was performed using the Metware Database (MWDB) based on retention time (RT) and daughter ion pair information of the detected substances. Lipid quantification was accomplished using the multiple reaction monitoring (MRM) mode of triple quadrupole mass spectrometry. In this mode, the quadrupole first screens the precursor ions (parent ions) of the target substance and excludes ions corresponding to other molecular weights to preliminarily eliminate interferences. The precursor ions are then induced to ionize by the collision chamber and break to form fragment ions, which are filtered through the triple quadrupole to select the characteristic fragment ions required, eliminating non-target ion interference for more accurate and reproducible quantification. After obtaining the lipid mass spectrometry data of different samples, peak area integration of all material peaks was performed, and quantitative analysis was carried out using the internal standard method.

### 2.6. Statistics

Unsupervised principal component analysis (PCA) was conducted using the prcomp function in R (www.r-project.org) accessed on 1 November 2023, with data scaled to unit variance prior to analysis. Hierarchical cluster analysis (HCA) and Pearson correlation coefficients (PCC) were calculated using the pheatmap package in R. HCA results were visualized as heatmaps with accompanying dendrograms, and PCC results were presented solely as heatmaps. Lipid differences between fresh and stored cassava flour were statistically analyzed by Partial Least Squares Discriminant Analysis (PLS-DA) and the heat map module in Metware Cloud, a free online platform for data analysis (https://cloud.metware.cn) accessed on 1 November 2023. This tool serves as a valuable resource for the preliminary screening of metabolic differences among different varieties or tissues [28]. For PLS-DA, the data were log-transformed (base 2) and mean-centered. VIP scores (VIP ≥ 2) and log2 fold changes (|log2FC| ≥ 1.0) were used to identify significantly regulated metabolites. The analysis, including score and permutation plots (200 permutations to prevent overfitting), was performed using the MetaboAnalystR package. Identified metabolites were annotated using the Kyoto Encyclopedia of Genes and Genomes (KEGG) Compound database (http://www.kegg.jp/kegg/compound, accessed on 1 November 2023) and mapped to metabolic pathways in the KEGG Pathway database (http://www.kegg.jp/kegg/pathway.html) accessed on 1 November 2023. Pathways with significantly regulated metabolites were further analyzed using metabolite set enrichment analysis (MSEA), with significance determined by hypergeometric test *p*-values. Statistical significance of differences in lipid levels during storage was assessed using one-way ANOVA in SPSS software (version 19.0), with *p*-values less than 0.05 considered statistically significant. The integrated peak area ratios of all detected lipids in SC9 and GR10 were used in the following calculation formula to obtain the absolute content data of the substance in the actual sample. The contents of these lipids in SC9 and GR10 with different storage time were calculated.
(1)$$X=\frac{0.001\times R\times C\times F\times V\times M}{m÷1000}$$
where X is the content of lipids in the sample (μg/g); R is the ratio of peak area of the substance to be measured to the peak area of the internal standard (area ratio); F is the internal standard correction factor for different types of substances; C is the concentration of internal standard (μmol/L); V is the sample extraction solution (μL); M is the molar mass; 1000 is the conversion factor; and m is the amount of sample weighed (g).

All classes of lipids (Appendix A) were analyzed for different storage times in SC9 and GR10 by using Origin 2018 software (Appendix A). Results were expressed as mean ± standard deviation, and all experiments were performed in triplicate.

## 3. Results and Discussion

### 3.1. Qualitative and Quantitative Analysis and Differences in Cassava Flour Lipids during Storage

In the positive ionization mode, the analysis detected signals corresponding to 470 lipid species across four classes and 17 subclasses, including ADGGA, DG, TG, LDGTS, etc., (Figure 1A,C). Notably, TG and LDGTS emerged as the most abundant lipid species. In contrast, the negative ionization mode detected 75 lipids belonging to two classes and 10 subclasses, such as FFA, LPA, PI, etc., (Figure 1B,D), with FFA and PI being the most prevalent. Upon filtering the detected lipids in both positive and negative ionization modes, a total of 545 lipids spanning five classes and 27 subclasses were identified across both cassava varieties (Table 1). The lipid profiles of the SC9 and GR10 cassava flour varieties were quantitatively compared by analyzing the content of individual lipid species within each subclass. SC9 (day 0) and GR10 (day 0) had essentially no significant differences in the individual lipids, except for a few, which may be due to variety differences. The data showed that some lipid content increased through storage, such as SP, and some DGTS in GR10, HexCer in SC9, and LDGTS in GR10, while the rest of the lipid content decreased. This phenomenon may be due to the hydrolysis of free fatty acids during lipid oxidation, and free fatty acids produce different oxidation products at the different stages of oxidation [29].

Sixteen types of FFA were observed in the two cassava flour varieties. The FFA content of SC9 increased from 468.01 ± 88.12 μg/g to 484.9 ± 44.58 μg/g at 180 days, while the FFA content of GR10 increased from 525.16 ± 59.36 μg/g to 717.51 ± 46.61 μg/g. After 540 days, the FFA values of the two cassava flour varieties decreased significantly (*p* < 0.05) to 323.03 ± 54.81 μg/g and 491.69 ± 87.51 μg/g, respectively. FFA concentrations in both cassava flours initially increased during storage due to the degradation of glycerolipids and glycerophospholipids [30], with a significant increase in FFA in GR10 (*p* < 0.05). When the amount of fatty acid produced was less than its degradation and volatilization amounts, the FFA values of both cassava flour varieties were significantly reduced (*p* < 0.05). This is consistent with changes in FFA during the storage of rice and buckwheat grains [31].

In this study, 31 species of DG were identified in two cassava varieties. The results showed a decrease in DG content during storage in both varieties, decreasing from 34.37 ± 5.26 μg/g and 48.06 ± 7.51 μg/g to 16.31 ± 2.28 μg/g and 36.07 ± 19.18 μg/g, respectively, suggesting that after 540 days, the reduction in DG is likely attributable to their involvement in the phospholipid biosynthetic pathway [32]. DG, a natural constituent of vegetable oils, fats, and plant tissues, is widely used in various industries [33].

In this study, 259 species of TG were identified in two cassava flour varieties. The TG content decreased significantly in both varieties during storage; SC9 decreased by 42.59% and GR10 by 62.50%, suggesting severe cell membrane damage under these conditions. TG and DG are the most susceptible to oxidation, differentiating fresh from stored cassava flour [34]. During storage, TG leaks out of damaged cell membranes, interacting with previously inactive but highly potent lipases and germs, which leads to the hydrolysis of TG. A similar situation was found in a related study by Milled Rice [35,36]. 

In this study, we identified six of LPA, 16 of LPC, nine of LPE, four of LPG, three of LPI, 12 of PA, three of PC, six of PE, eight of PG, 14 of PI, two of PMeOH, and three of PS. The comparison of lipid contents revealed that storage increased the levels of LPA, LPC, and LPG, while PC and PS decreased in both varieties. Degradation products such as PE, PI, and PMeOH initially increased and then decreased in SC9 and decreased in GR10, whereas PI and PG both decreased initially and then increased. Glycerophospholipids play numerous roles in living systems. As major components of biological membranes, they significantly influence membrane stability and permeability [37].

Two CoQ species were identified in this study. Storage had no significant effect on the CoQ content of SC9, whereas it significantly reduced the CoQ content of GR10 (*p* < 0.05) by approximately 96.92%. CoQ, a redox-active lipid, plays a central role in cellular homeostasis. It is a cofactor for several mitochondrial dehydrogenases and is involved in lipid, amino acid, and nucleotide metabolism as well as sulfide detoxification. It also regulates apoptosis and iron apoptosis [38]. CoQ deficiency in plants impacts their growth and development. CoQ biosynthesis has been studied mainly in yeast and human cells, so our knowledge of the diversity of CoQ biosynthesis pathways in eukaryotes is very limited, whereas in Arabidopsis, there is a similar study of CoQ substances [39].

The Cer content in both cassava flours remained stable during storage, while Cert and HexCer content significantly decreased in the GR10 variety (*p* < 0.05), decreasing by 22.83% and 34.10%, respectively; this suggests that storage led to a decrease in Cert and HexCer contents specifically in GR10, potentially contributing to mitochondrial dysfunction and apoptosis, thus impacting the storage quality of GR10 [40]. Sphingolipids are bioactive lipids present in cell membranes and are crucial for signal transduction processes [41]. Cer, Cert, HexCer, and SPH are integral components with significant roles in various cellular processes, serving as fundamental structural compounds within cell membranes and exerting essential functions in life processes [42,43]. In particular, Cer is involved in mitochondria-mediated apoptosis [40].

Sixteen species of DGDG, 14 species of MGDG, and 11 species of SQDG were observed in both cassava varieties. Glycolipids (MGDG, DGDG, and SQDG) did not change significantly in both varieties of cassava flour, suggesting that storage has no significant effect on the saccharolipids-associated fraction of cassava flour cell membranes. Saccharolipids, such as DGDG, SQDG, and MGDG, are significant structural constituents of cellular membranes [44].

In this study, eight of DGTS and seven of LDGTS were identified. Storage had no significant effect on the DGTS content in both varieties, while there was a significant decrease in the LDGTS content of SC9 and a significant increase in the content of GR10 (*p* < 0.05), which decreased by 46.28% and increased by 55.63%, respectively. DGTS and LDGTS improve the quality and anti-atherosclerotic function of high-density lipoproteins. There are related metabolites studied in Taraxacum officinale [45]. 

Lipid changes in flour during storage have been documented [46,47]. The changes are similar to the results of this study. Further studies are needed, as essentially no scientific studies directly confirm the lipidomic profile of cassava flour during storage.

### 3.2. Significant Results of Differences in Lipids in Fresh and Stored Cassava Flour

This study analyzed the impact of storage duration on the lipidomic profiles of two cassava flours, SC9 and GR10, using the PLS-DA method. The cassava flours were categorized into two groups: fresh cassava flour (day 0) and stored cassava flour (days 180 and 540).

As shown in the lipid species score plot (Figure 2A), the fresh and stored SC9 samples were distinctly separated based on the first two principal components, which accounted for a cumulative contribution of 89.5%. The VIP scores of lipids contributing to the differentiation of SC9 were estimated. As shown in Figure 2B, two lipid species variables (MGDG, DGTS) were considered significant contributors (VIP score > 2). These lipid subclasses can be utilized in future studies to distinguish between fresh and stored cassava flour (SC9). 

The heat map clearly visualizes the lipid distribution in fresh and stored cassava flour. As shown in Figure 2C, it supports the analyses presented in Appendix A and Table 1. The score plot of individual lipids (Figure 2D) demonstrates a distinct separation between fresh and stored SC9, based on the first two principal components, which account for 88.9% of the variance. The VIP scores for each lipid fraction’s influence on malignant weight were also assessed. Figure 2E highlights 11 lipid variables with significant contributions (VIP score > 2), including four DGDGs, three DGs, one FFA, one MGDG, one DGTS, and one SQDG. These lipids can also be used to distinguish between fresh and stored SC9.

From Figure 3A, fresh (day 0) and stored GR10 (days 180 and 540) samples can be clearly classified based on the first two principal components, which accounted for a cumulative contribution of 73.55%. As shown in Figure 3B, two lipid species variables, CoQ and PE, were considered to have significant contributions (VIP score > 2). These lipid species can be utilized to differentiate between fresh and stored GR10 in future studies. 

The heat map in Figure 3C also confirms the analysis of Appendix A and Table 1. An analysis of the individual lipids in both fresh and stored cassava flour (GR10) demonstrated that these samples could be clearly distinguished using the first two principal components, which together account for 81.88% of the variance (Figure 3D). As observed in Figure 3E, a total of 39 individual lipid variables, including 37 TGs, one CoQ, and one DG, were considered to have significant contributions (VIP score > 2). Individual lipid analyses can also differentiate between fresh and stored GR10.

In summary, there are 50 significantly differentiated lipids in the two varieties of cassava flour during storage, most of which are GL, which is similar to the results of the study on lipid changes during the storage of wheat flour [48].

### 3.3. Lipid Metabolism Pathways during Cassava Flour Storage

To elucidate the changes in lipid metabolic pathways during cassava flour storage, 50 significantly differentiated lipids (from the combined SC9 and GR10 varieties) were mapped to the KEGG database to obtain their ID information. 

To identify the different metabolites, KEGG enrichment maps were created (Figure 4A). Enrichment factors were calculated as the ratio of total metabolites identified by pathway identification to the differentially expressed total metabolites in the pathway of interest, with higher values indicating greater accumulation. Figure 4B presents the calculated hypergeometric test *p*-values. In the KEGG enrichment plot for differential metabolites, the horizontal coordinate represents the enrichment factor for each pathway, the vertical coordinate represents the pathway name, and the color of the dots indicates the *p*-value. The size of each dot represents the number of differentially enriched metabolites. 

The entirety of the KEGG annotation results of differentially significant metabolites were classified based on KEGG pathways, as shown in Figure 4C [49]. These significant lipids participated in seven metabolic pathways: metabolic pathways, inositol phosphate metabolism, glycerophospholipid metabolism, glycerolipid metabolism, the biosynthesis of unsaturated fatty acids, the biosynthesis of secondary metabolites, and the phosphatidylinositol signaling system. Among these, metabolic pathways were the most significantly different, followed by glycerolipid metabolism (Figure 4). DG and TG were involved in both of these lipid metabolism pathways.

According to the KEGG pathway analysis, the glycerolipid metabolism pathway involves the hydrolysis of glycerophospholipids, diglycerides, and triglycerides [50,51,52], resulting in a lower content of diglycerides and triglycerides in stored SC9 and GR10 compared to fresh samples. Diglycerides and triglycerides are also involved in metabolic pathways and the biosynthesis of secondary metabolites. We believe that the KEGG pathway analyses of the differing lipids in fresh and stored cassava flour provide valuable insights into the changes and functions of cassava lipids over storage periods. However, the oxidative degradation products of fatty acids and phenolic acids need further investigation to fully explain and confirm these lipid metabolic pathways.

### 3.4. Possible Mechanisms of the Glyceride Metabolic Pathway in Cassava Flour during Storage

According to the KEGG pathway analysis, based on the changes in lipid distribution and the relevant literature, we propose a possible mechanism for glyceride metabolism in cassava flour during storage (Figure 5). During storage, sn-Glycerol 3-phosphate reacts with Acyl-CoA and glycerol-3-phosphate O-acyltransferase 1/2 to form 1-Acyl-sn-glycerol 3-phosphate. This intermediate continues to react with Acyl-CoA and 1-acyl-sn-glycerol-3-phosphate acyltransferase to form 1,2-Diacyl-sn-glycerol 3-phosphate. Subsequently, 1,2-Diacyl-sn-glycerol 3-phosphate reacts with phosphatidate phosphatase to form 1,2-Diacyl-sn-glycerol, which then reacts with Acyl-CoA and diacylglycerol O-acyltransferase 1 to form Triacylglycerol (TG) through the Kennedy pathway, a major pathway for TG biosynthesis in most tissues or organisms [53]. 

TG is reduced via the acylglycerol degradation pathway, reacting with triacylglycerol lipase to form 1,2-Diacyl-sn-glycerol. This then reacts with phosphatidate phosphatase to form 1-Acylglycerol, which in turn, through acylglycerol lipase, produces glycerol and fatty acids. The degradation of fatty acids explains the reduction of free fatty acids in cassava flour during storage. TG biosynthesis is closely related to human health and significantly impacts the economic value of oilseed crops and oil-producing microorganisms, making the study of lipid metabolism crucial.

Glycerol from TG degradation, a polyol produced during the metabolism of glycerolipids such as glucose, pyruvate, and TG, is a common cellular metabolite found in living organisms. It is also a direct end product of plant photosynthesis and can be obtained from the breakdown of storage lipids [54]. 

TG synthesis and degradation are affected by water activity, temperature, time, and pH [55]. The hydrolysis of triacylglycerols leads to rancidity, producing free fatty acids and glycerol. Acid rancidity produces reactive oxygen species (ROS), which subsequently cause unpleasant tastes and odors in edible food [56]. Therefore, regulating storage conditions to prevent rancidity is a key concern for the future storage of cassava flour.

## 4. Conclusions

In this study, the lipidomic profiles and changes in SC9 and GR10 cassava flour during storage (0, 180, and 540 days) were analyzed using LC-ESI-MS/MS. A total of 545 lipids from five classes and 27 subclasses were identified. Bioinformatics analysis was employed to identify 50 significantly different lipids during the storage of cassava flour. The identified significantly different lipids (DG and TG) and related metabolic pathway (glycerolipid metabolism pathway) in this study can serve as a reference for the lipid rancidity of cassava flour, which provides a practical method for the rancidity process of cassava flour that was investigated, and it can evaluate the storage quality of cassava flour.

## Figures and Tables

**Figure 1 foods-13-03039-f001:**
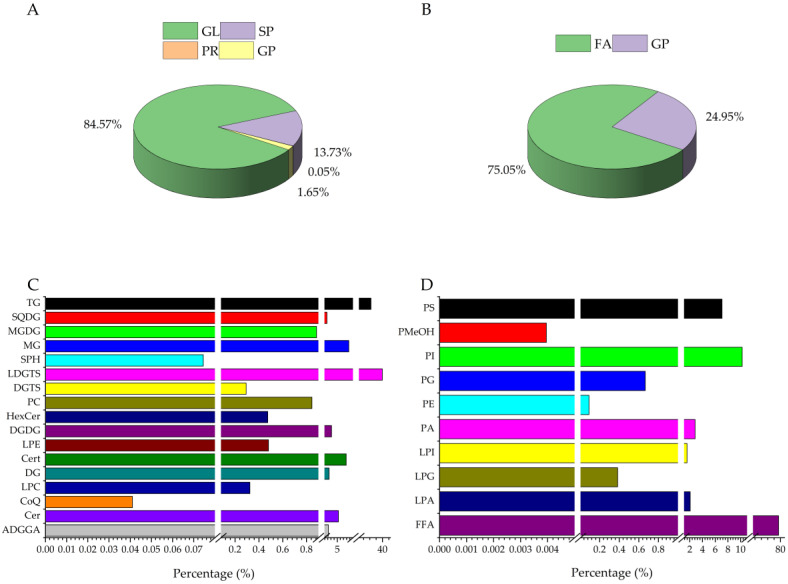
Percentage of two cassava flour lipid classes in positive and negative ionization mode (**A**,**B**), Percentage of two cassava flour lipid subclasses in positive and negative ionization mode (**C**,**D**).

**Figure 2 foods-13-03039-f002:**
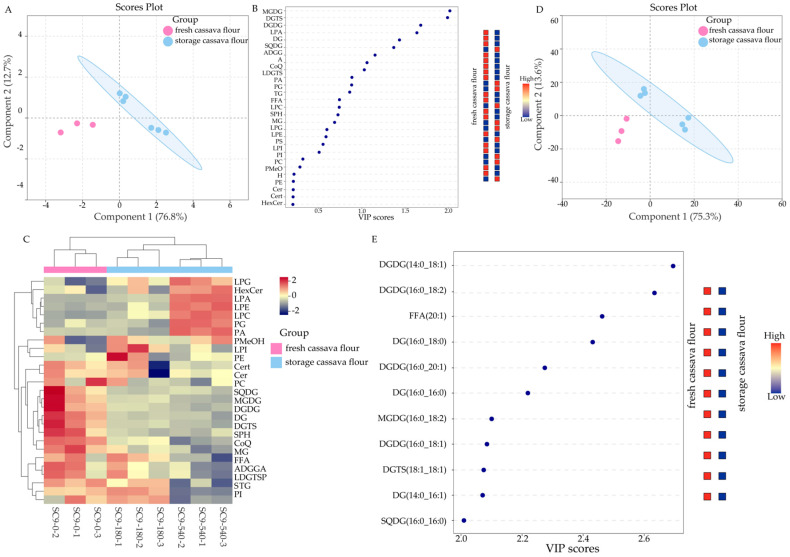
Metware Cloud statistical analysis of lipids during SC9 storage. The lipid species score plot of PLS-DA (**A**), VIP scores of lipid species in PLS-DA (**B**), heat map of different lipid subclasses (**C**), individual lipids score plot of PLS-DA (**D**), and the VIP scores of individual lipids in PLS-DA (**E**). Colored boxes on the right indicate the relative concentration of the corresponding lipid. Red indicates high and blue indicates low.

**Figure 3 foods-13-03039-f003:**
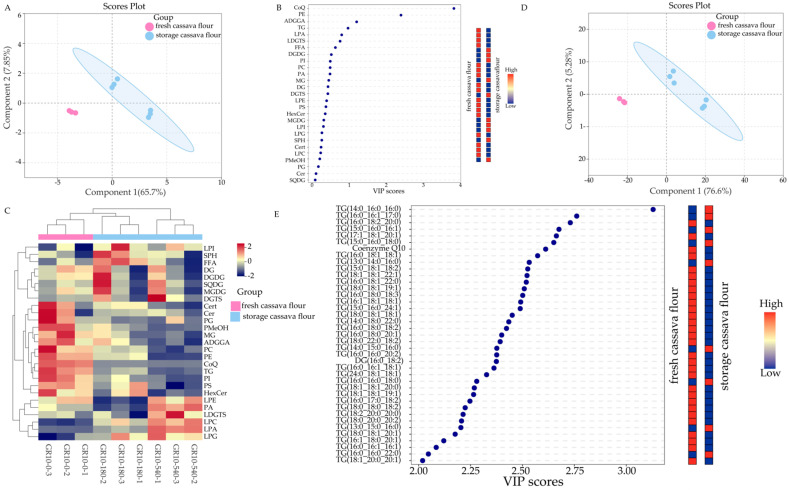
Metware Cloud statistical analysis of lipids during GR10 storage. The lipid species score plot of PLS-DA (**A**), VIP scores of lipid species in PLS-DA (**B**), heat map of different lipid subclasses (**C**), individual lipids score plot of PLS-DA (**D**), and the VIP scores of individual lipids in PLS-DA (**E**). Colored boxes on the right indicate the relative concentration of the corresponding lipid. Red indicates high and blue indicates low.

**Figure 4 foods-13-03039-f004:**
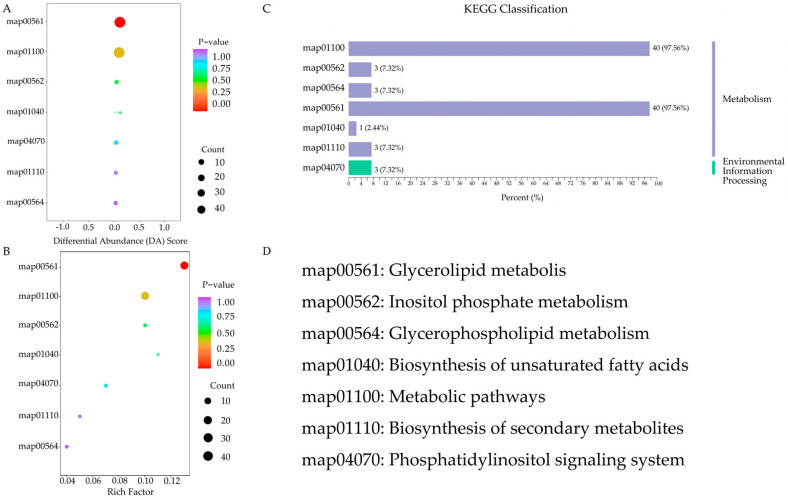
(**A**) Differential abundance score of the metabolic pathways. (**B**) KEGG enrichment map. (**C**) KEGG enrichment analysis of the lipid metabolites. (**D**) Metabolic pathway entry number and corresponding name. (KEGG pathway analysis using Metware Cloud also highlighted significant differences in lipid profiles between fresh cassava flour (day 0) and cassava flour stored for 180 and 540 days. Differential abundance (DA) scores were calculated using pathway-based metabolic change studies, ranging from −1 to 1. A score of −1 indicated a down-regulated trend, while a score of 1 indicated an up-regulated trend).

**Figure 5 foods-13-03039-f005:**
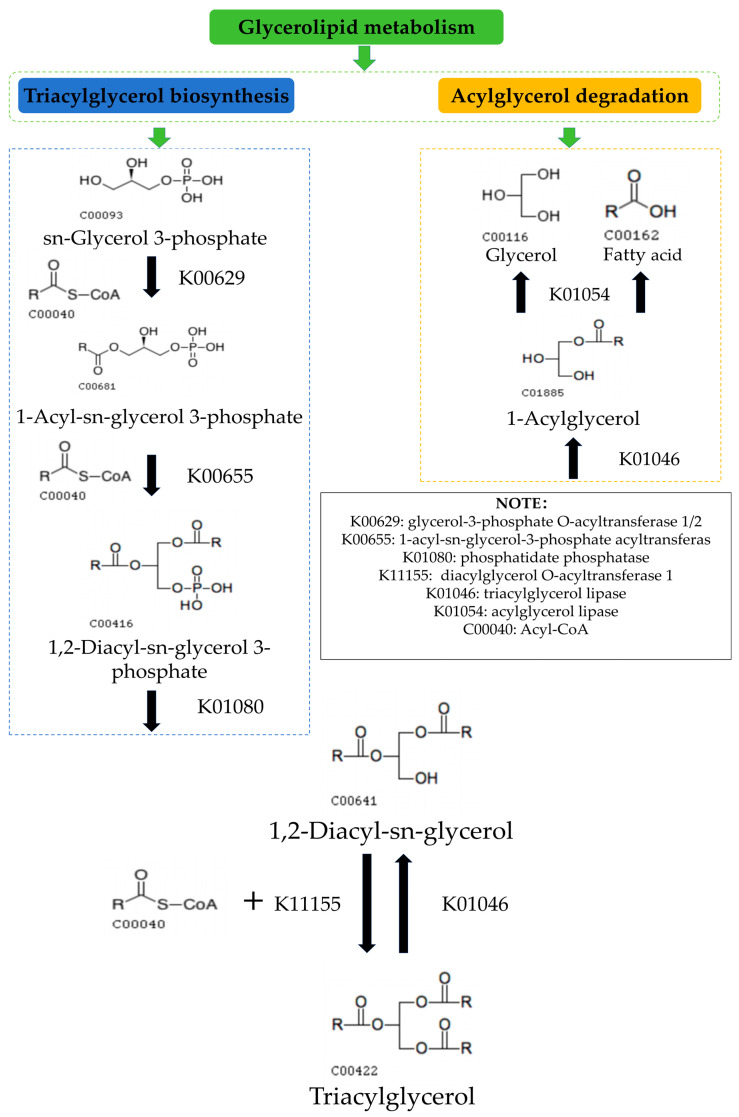
The proposed mechanism of glyceride metabolism in cassava flour during storage.

**Table 1 foods-13-03039-t001:** Lipid content in cassava flour during storage. (*n* = 3).

Lipid		Content of Lipids (μg/g)					
Class I	Class II	SC9 (Day 0)	SC9 (Day 180)	SC9 (Day 540)	GR10 (Day 0)	GR10 (Day 180)	GR10 (Day 540)
GL	ADGGA	32.37 ± 9.06 ^abc^	25.05 ± 6.57 ^ab^	15.59 ± 5.69 ^a^	50.25 ± 13.31 ^c^	35.69 ± 13.80 ^bc^	14.26 ± 5.97 ^a^
	DG	34.37 ± 5.26 ^ab^	17.89 ± 0.56 ^a^	16.31 ± 2.28 ^a^	48.06 ± 7.51 ^b^	39.08 ± 21.97 ^ab^	36.07 ± 19.18 ^ab^
	DGDG	42.90 ± 13.11 ^ab^	20.35 ± 1.16 ^ab^	15.98 ± 1.51 ^a^	66.24 ± 8.92 ^b^	62.91 ± 50.17 ^ab^	45.52 ± 29.34 ^ab^
	DGTS	5.07 ± 1.32 ^ab^	2.10 ± 0.11 ^ab^	1.62 ± 0.26 ^a^	3.73 ± 0.47 ^ab^	4.31 ± 3.37 ^ab^	6.62 ± 4.69 ^b^
	LDGTS	749.41 ± 197.12 ^c^	629.13 ± 175.44 ^abc^	402.61 ± 135.00 ^ab^	414.60 ± 75.37 ^ab^	339.79 ± 147.57 ^a^	645.23 ± 160.55 ^bc^
	MG	361.47 ± 38.28 ^b^	278.51 ± 26.11 ^a^	233.50 ± 27.64 ^a^	467.43 ± 59.14 ^c^	358.59 ± 59.74 ^b^	292.58 ± 24.36 ^ab^
	MGDG	14.94 ± 5.18 ^a^	6.32 ± 0.47 ^a^	4.22 ± 0.47 ^a^	23.14 ± 5.80 ^a^	23.22 ± 17.75 ^a^	20.37 ± 14.53 ^a^
	DGTS	6.83 ± 1.77 ^ab^	2.84 ± 0.15 ^ab^	2.18 ± 0.36 ^a^	5.02 ± 0.63 ^ab^	5.81 ± 4.55 ^ab^	8.94 ± 6.35 ^b^
	SQDG	25.15 ± 8.42 ^a^	12.35 ± 0.57 ^a^	12.98 ± 1.45 ^a^	25.04 ± 4.71 ^a^	31.60 ± 26.57 ^a^	29.50 ± 19.13 ^a^
	TG	308.88 ± 11.10 ^b^	343.69 ± 17.17 ^b^	177.34 ± 10.00 ^a^	626.65 ± 74.23 ^c^	343.81 ± 39.52 ^b^	235.01 ± 15.19 ^a^
SP	Cer	75.77 ± 3.38 ^a^	69.03 ± 12.58 ^a^	71.22 ± 1.43 ^a^	76.89 ± 13.74 ^a^	70.20 ± 0.64 ^a^	68.79 ± 4.57 ^a^
	Cert	111.11 ± 5.23 ^ab^	102.24 ± 18.18 ^ab^	99.61 ± 1.25 ^ab^	116.19 ± 20.79 ^b^	108.67 ± 2.80 ^ab^	89.66 ± 5.21 ^a^
	HexCer	5.94 ± 0.60 ^ab^	6.06 ± 0.61 ^ab^	7.03 ± 0.16 ^b^	7.42 ± 1.15 ^b^	6.60 ± 1.38 ^b^	4.89 ± 0.36 ^a^
	SPH	1.13 ± 0.15 ^b^	0.80 ± 0.03 ^a^	0.80 ± 0.04 ^a^	1.02 ± 0.07 ^b^	1.17 ± 0.09 ^b^	1.04 ± 0.09 ^b^
PR	CoQ	0.18 ± 0.01 ^ab^	0.12 ± 0.01 ^ab^	0.09 ± 0.01 ^ab^	2.60 ± 0.17 ^c^	0.21 ± 0.01 ^b^	0.08 ± 0.01 ^a^
FA	FFA	468.01 ± 88.12 ^b^	484.91 ± 44.58 ^b^	323.03 ± 54.81 ^a^	525.16 ± 59.36 ^b^	717.51 ± 46.61 ^c^	491.69 ± 87.51 ^b^
GP	LPA	7.97 ± 0.04 ^a^	10.59 ± 0.50 ^b^	28.01 ± 0.61 ^d^	7.71 ± 0.13 ^a^	10.09 ± 0.78 ^b^	18.99 ± 0.64 ^c^
	LPC	3.84 ± 0.11 ^bc^	4.65 ± 0.33 ^d^	6.65 ± 0.16 ^e^	3.15 ± 0.05 ^a^	3.65 ± 0.16 ^b^	4.00 ± 0.11 ^c^
	LPE	5.87 ± 0.15 ^b^	6.30 ± 0.56 ^b^	9.10 ± 0.26 ^c^	6.05 ± 0.26 ^b^	4.91 ± 0.11 ^a^	6.19 ± 0.33 ^b^
	LPG	2.18 ± 0.25 ^a^	2.71 ± 0.16 ^bc^	3.05 ± 0.18 ^c^	2.15 ± 0.20 ^a^	2.59 ± 0.23 ^b^	2.71 ± 0.21 ^bc^
	LPI	9.46 ± 0.96 ^a^	11.23 ± 0.84 ^ab^	9.39 ± 0.62 ^a^	9.89 ± 1.34 ^a^	12.10 ± 1.46 ^b^	11.24 ± 0.88 ^ab^
	PA	16.39 ± 0.25 ^bc^	14.65 ± 1.42 ^a^	28.25 ± 0.97 ^e^	17.97 ± 0.88 ^c^	14.94 ± 0.71 ^ab^	21.86 ± 0.80 ^d^
	PC	11.73 ± 1.62 ^b^	10.57 ± 1.17 ^ab^	9.83 ± 0.83 ^ab^	15.12 ± 2.42 ^c^	11.62 ± 0.40 ^b^	8.78 ± 0.75 ^a^
	PE	0.31 ± 0.01 ^a^	0.34 ± 0.06 ^a^	0.31 ± 0.01 ^a^	1.66 ± 0.27 ^c^	0.95 ± 0.09 ^b^	0.12 ± 0.02 ^a^
	PG	3.75 ± 0.48 ^ab^	3.47 ± 0.47 ^a^	6.54 ± 0.23 ^d^	4.67 ± 0.43 ^c^	4.11 ± 0.11 ^abc^	4.19 ± 0.32 ^bc^
	PI	58.62 ± 7.66 ^ab^	62.49 ± 0.95 ^b^	46.80 ± 2.45 ^a^	103.63 ± 10.90 ^d^	76.25 ± 8.60 ^c^	62.74 ± 4.51 ^b^
	PMeOH	0.02 ± 0.003 ^a^	0.03 ± 0.001 ^ab^	0.03 ± 0.001 ^ab^	0.03 ± 0.003 ^b^	0.03 ± 0.002 ^a^	0.02 ± 0.003 ^a^
	PS	49.90 ± 2.25 ^c^	46.06 ± 2.86 ^bc^	32.85 ± 5.32 ^a^	60.89 ± 3.08 ^d^	53.80 ± 8.14 ^cd^	39.93 ± 5.27 ^ab^

Mean ± standard deviation of triplicate measurements. Distinct superscript letters in the same row indicate significant differences (*p* < 0.05).

## Data Availability

The original contributions presented in the study are included in the article/Appendix A, further inquiries can be directed to the corresponding author.

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
