# Peer review of "Lipidomic Profiling and Storage-Induced Changes in Cassava Flour Using LC-MS/MS"

_foods, 2024, doi:10.3390/foods13193039_

Round 1
Reviewer 1 Report
Comments and Suggestions for Authors
Comments and recommendations to the authors:
1. In Materials and methods section (page 3, line 100): It is not described which harvest of the cassava roots was and where they were taken from (origin).
2. In Materials and methods section (page 3, line 101): It is mentioned that “The flour was stored in a sample library at room temperature (25-30°C) with humidity controlled at 60%±5%.” How these conditions (temperature and humidity) were chosen?
3. In Results section (page 5, line 208): The authors declare that “The data showed that most lipid content increased through storage, such as……”. I recommend to describe the possible reason for this increase.
4. In Results section (page 6, line 262): It is mentioned that “Lipid changes in rice and flour during storage have been documented”. Is there any comparison between rice and cassava?
5. The discussion of the results and their interpretation is poor.
6. The conclusion section is missing. Probably, the name of the fourth part “Discussion” can be changed in “Conclusion”.

Reviewer 2 Report
Comments and Suggestions for Authors
Abstract and Introduction – I would suggest supplementing with a specific purpose for this study.
Line 81-90 – does not really fit into the introduction.
Line 100 – More precisely, how the drying took place.
Line 101 – What kind of equipment do you grind into flour? Was the ground flour stored without packaging? Were they stored packaged? What material was the packaging? How was the flour packaged?
All equipment used must specify make, manufacturer, country.
Line 165 – When mentioning an abbreviation for the first time, it is necessary to decipher it.
Results – I would suggest that the authors revise the text and move the section to the "Discussion" section. Is there any way to compare the data obtained between the "SC9(day 0)" and "GR10(day 0)" samples?
Line 237-242 – Why is this text underlined?
The "Discussion" section needs to be revised and supplemented.
Where are the conclusions for this particular study?
Reviewer 3 Report
Comments and Suggestions for Authors
The research of the current manuscript concerns the evaluation of the lipidomic profiling in Cassava flour during a storage period using LC-MS/MS.
In my opinion the manuscript needs a deep revision.
Introduction.
It is not essential and it doesn’t focus on the manuscript target. Furthermore, it is unbalanced toward the lipidomic studies whereas the lipid oxidation of flour in general and cassava flour, in particular was not treated at all. This is an uncorrected way to write an Introduction. In my opinion, this section should be written bringing the author to know the current status of the research and the issues related to the topic.
Material and methods
Authors completely forgot to describe the real time shelf-life storage. They only reported some confused information about that, in the Introduction. This is a huge error which it cannot be admitted by scientific researchers.
Moreover, I suggest a sensory analysis to evaluate the rancidity defect evolution which strongly compromise the consumers acceptance; in addition, sensory analysis results could be matched by a proper statical approach (i.e. PLS or PLS-DA) with instrumental analysis in order to correlate data and find, among the chemical substances analyzed, those ones responsible for the rancid defect (already correlated by other authors with fatty oxidation products in foods).
Results and Discussion
Results are not well illustrated. Authors should report data or percentage decrease. Authors, in fact limited the results description by this statement “not significant/significant”.
Furthermore, results discussion lacked of a proper discussion of data and a relative comparison with other scientific experimentation capable to corroborate or contrast own results.
The phrases in lines 237 and 242 were underlined…why???
I would like underline again that this scientific research could have more importance if supported by a proper descriptive quantitative sensory analysis.
Tables and Figures.
Authors forgot to report the acronym significance of the chemical compounds in all the legends.
